# Winter Wheat Grain Quality, Zinc and Iron Concentration Affected by a Combined Foliar Spray of Zinc and Iron Fertilizers

**Etienne Niyigaba, Angelique Twizerimana, Innocent Mugenzi, Wansim Aboubakar Ngnadong, Yu Ping Ye, Bang Mo Wu and Jiang Bo Hai \***

Department of crop cultivation and farming system, College of Agronomy, Northwest A&F University, Yangling, Shaanxi 712100, China; niyigabat@gmail.com (E.N.); atwizerimana@gmail.com (A.T.); mugenziinn@gmail.com (I.M.); aboubakar777@yahoo.fr (W.A.N.); 2012010065@nwafu.edu.cn (Y.P.Y.); wu_bangmo@126.com (B.M.W.)

\* Correspondence: haijiangbo@126.com; Tel.: +86-13389221092

**Abstract:** Wheat (*Triticum aestivum* L.) is one of the main foods globally. Nutrition problems associated with Zinc and Iron deficiency affect more than two billion individuals. Biofortification is a strategy believed to be sustainable, economical and easily implemented. This study evaluated the effect of combined Zn and Fe applied as foliar fertilizer to winter wheat on grain yield, quality, Zn and Fe concentration in the grains. Results showed that treatments containing high Fe increased the yield. Grain crude fat content remained unaffected. Crude fiber was enhanced up to three-fold by 60% Zn + 40% Fe$_{5.5}$ (5.5 kg ha$^{-1}$ of 60% Zn + 40% Fe). Moreover, 80% Zn + 20% Fe$_{5.5}$ (5.5 kg ha$^{-1}$ of 80% Zn + 20% Fe) was the best combination for increasing crude protein. Zinc applied alone enhanced Zn concentration in grain. In addition, Fe was slightly improved by an application of Zn and Fe in the first year, but a greater increase was observed in the second year, where 100% Fe$_{13}$ (13 kg ha$^{-1}$ of 100% Fe) was the best in improving Fe in grain. Foliar application of Zn and Fe is a practical approach to increase Zn and Fe concentration, and to improve the quality of wheat grains.

**Keywords:** combined zinc and iron fertilizers; foliar spray; agronomic biofortification; wheat

## 1. Introduction

Wheat (*Triticum aestivum* L.) is among the top cereal crops cultivated in the world, together with maize and rice. They contribute to edible dry matter, and daily calorie intake, up to 28% and 60%, respectively, in developing countries (GNI < $12,055) [1–3]. However, in developing countries, nutrition deficiency is a serious problem associated with poor diet. Food and nutrient intake constitute the basis of life; people are dying en masse due to a lack of sufficient nutrients [4]. This problem resulted in the deaths of about 30 million people in 2003 in developing countries with poor resources [5,6]. As reported by Graham [4], every second, one person dies of disease related to diet. Moreover, Fe deficiency affects more than two billion individuals, or one in three people globally [7,8], while about 30% of people in developing countries and 10% of Americans and Canadians are Zn deficient [9]. The World Health Organization estimates that approximately 25% of the world's population suffers from anemia [10], and that Fe deficiency anemia led to the loss of over 46,000 disability-adjusted life years (DALYs) in 2010 alone [11]. An estimated 17.3% of people worldwide are at risk of inadequate Zn intake, and Zn deficiency leads to estimated annual deaths of 433,000 children under the age of 5 [12].

Recently, it was reported that in the UK, the Zn intake of about a quarter of adolescents is below the lower reference nutrient intake (LRNI) and the Fe intake of over half of all adolescent girls and over a quarter of adult women is below the LRNI [13,14]. Nutrient-deficient food and diet related problems

are the main cause of death on Earth, and this can be prevented by supplying nutrients in a sustainable manner, and finding solutions to malnutrition [4].

The human body needs 51 essential nutrients, and a short supply or lack of even one of these can cause metabolic problems resulting in poor health, sickness, and economic and social costs to the community [4,7,15,16]. These nutrients are supplied only from agricultural products (excluding water and oxygen). The green revolution has boosted crop yield, and has prevented people from starving in many countries through the high production of cereal crops (wheat, maize and rice) [4]. However, the cereal monoculture system has enhanced the micronutrient deficiency problem by eliminating diverse cropping systems that provided micronutrient-rich food for poor communities [17]. In addition, breeding of high-yielding crops has caused a significant "dilution effect" of essential nutrients, such as Zn and Fe [18–20]; Cakmak reported that modern breeds have high yield, but their respective wild types present from two to three-fold more Zn [21,22]. However, wheat is inherently low in Zn, and its consumption in rural areas is likely to increase to more than 70%, which amplifies the magnitude of nutrient deficiency in communities with poor resources [23].

Cereals contribute up to 60% and 52% of the daily micronutrient requirement for Fe and Zn, respectively [24]. Therefore, it is important to focus on the important food crops that maintain human life in most countries. Several initiatives are now in place to deeply study and contribute to the nutritional quality of food. For instance HarvestPlus, the Biofortification Challenge Program established by the Consultative Group on International Agricultural Research (CGIAR), is focusing on enhancing major staple crops with Fe, Zn and β-carotene [4,18].

Biofortification is a process of developing high micronutrient food crops by traditional breeding or modern biotechnology. Biofortification of staple crops has had major developments recently: Orange-flesh sweet potato, with high β-carotene (over 200 μg/g), beans with 50–70% more Fe, golden rice with 37 μg/g carotenoid, of which 31μg/g is β-carotene, have been bred to date. Moreover, research is taking place to increase genetically the Fe in rice grain endosperm from the aleurone layer, which is removed when producing polished rice [25]. It was reported that the Zn concentration of most cultivated areas range between 20 and 35 mg kg$^{-1}$, and it could be far below this when wheat is grown on zinc-poor soils [4,18,26,27]. These concentrations cannot meet daily Zn requirements to reach the target range for human health, which is 40–50 mg kg$^{-1}$ [4,27,28].

In wheat grains, the bioavailability of Zn is about 25%, while that of Fe is assumed to be 5%. The bioavailability of Fe and Zn is associated with the presence of antinutrients, such as phytate and a lack of promoter substances in grains [7,29]. Any breeding or biofortification program should consider increasing not only the quantity of micronutrients, but also their bioavailability [22].

Some nutrients facilitate the uptake and remobilization of Fe and Zn in plants. For instance, the combination of nitrogen fertilizer with Zn and Fe applied in soil or on leaves increases both the yield and the uptake of these elements [27,30]. In wheat, the translocation of Zn and Fe from flag leaves to grains is also facilitated by metal-chelating compounds, such as 2-deoxymugineic acid (DMA) [31]. At a high N rate, a great part of total shoot Zn and Fe, nearly 80% and 60%, respectively, was found in grains, highlighting the role of N in supporting the movement of Zn and Fe in wheat [32]. Similarly, Erenoglu [33] illustrated that biofortification of food crops needs to take into consideration the critical role of nitrogen in the uptake and accumulation of Zn. Furthermore, HarvestPlus reported a positive effect of Nitrogen upon the concentration of Zn and Fe in wheat. This relationship allows the increase of multiple micronutrients simultaneously [22]. The role of nitrogen in facilitating the uptake, transport, translocation, and deposits of micronutrients, especially Zn and Fe in cereal grains, has been extensively studied, and reports are available [8,32–36]. Moreover, S (sulfur) is another element that was repeatedly reported in assisting Fe and Zn metabolism in plants. The plant's ability to absorb and accumulate Fe was proven to be dependent upon S presence in the growing medium in cereal crops [37,38].

The release of phytosiderophores that help Fe uptake by the roots has a positive correlation with S accumulation, and this process can increase the Fe use efficiency of roots, hence alleviating the effect of

Fe deficiency [38,39]. Astolfi et al. reported an important finding in the biofortification of wheat grains with Zn that, irrespective of the S application level, the Zn concentration in durum seed remained higher in Fe-deficient plants, compared to control plants, and the study revealed that S fertilization can alleviate the Fe shortage stress without harmful effects on grain quality [40]. Several available resources demonstrate the S and Fe interaction in plants [37,38,40–43].

There are many possible strategies to improve micronutrient intake in the human diet, including dietary diversification, mineral supplementation, post-harvest food fortification and biofortification [44]. Plant breeding (e.g., genetic biofortification) and application of Zn and Fe fertilizers (e.g., agronomic biofortification), are two important agricultural tools to improve the grain concentration of Zn and Fe [22]. Even though genetic engineering opens more doors to increase dramatically the bioavailability of Zn and Fe in grains, its acceptability by consumers and regulatory bodies is very limited, and genetically modified crop cultivation and marketing are not likely to be relaxed in the near future [13]. Agronomic biofortification is achieved by applying micronutrients to the soil and/or directly to the leaves of the crop [45]. In contrast to genetic engineering, agronomic biofortification is potentially more sustainable, more economical, and more easily implemented than other strategies [23,46,47].

Foliar application of nutrients is an important crop management strategy to maximize crop yields and concentrations of micronutrients in edible parts. Several studies have demonstrated that foliar application of micronutrients, including Zn and Fe, showed good behavior in increasing their concentration in wheat grain [27,48–52]. However, little is known about the effect of combined Zn and Fe on the quality of this wheat grain. Therefore, this study was conducted to closely assess the combined effect of Zn and Fe applied as foliar fertilizer at different growth stages of winter wheat on crop performance, grain yield and grain quality.

## 2. Materials and Methods

### 2.1. Field Location

This experiment was conducted during the winter wheat cropping seasons of 2016–2017 and 2017–2018 at the Doukou Wheat and Maize Demonstration Research Station (108°52′ E, 34°36′ N) of Northwest A&F University, Shaanxi Province, China. The site's climate is classified as semi-humid. The average temperature during the trials period was 9.5 °C and 10.4 °C, during 2016–2017 and 2017–2018 season, respectively (Figure 1). The average precipitation was 16.4 mm and 11.6 mm, during 2016–2017 and 2017–2018 season, respectively (Figure 1). In addition, the station's average annual solar radiation is 247.35 kJ cm$^{-2}$, the average annual sunshine duration is 2271.6 h, and the sunshine percentage is 51%. The average annual frost-free period is about 220 days, while the average annual precipitation is 595 mm, and the average annual land surface evaporation is 417.6 mm. The soil used is classified as Earth-Cumuli Orthic anthrosol, with total N of 151.00 mg kg$^{-1}$, available K of 26.97 mg kg$^{-1}$, available P of 198.54 mg kg$^{-1}$, pH of 8.04 (alkali), extractable Zn of 1.35 mg kg$^{-1}$ and extractable Fe of 7.87 mg kg$^{-1}$. The soil used is classified as having high Zn and Fe content [53].

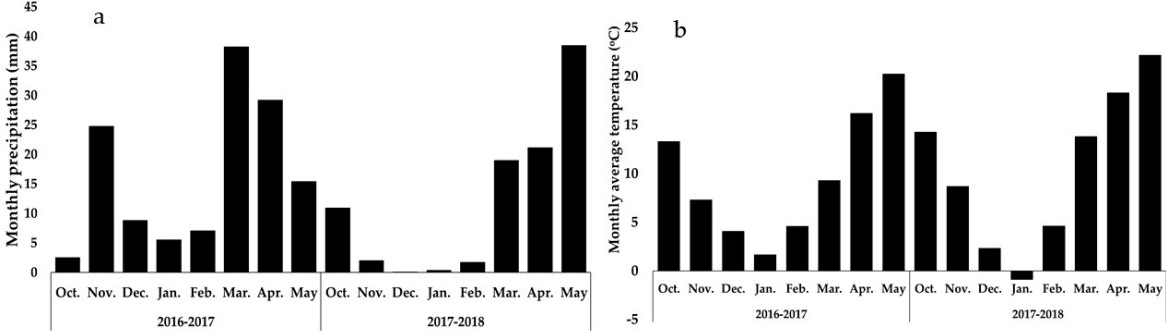

**Figure 1.** Meteorological data during both cropping seasons: (**a**) Monthly precipitation, (**b**) monthly average temperature.

## 2.2. Experimental Design and Treatments

The experimental design consisted of split-plots in a randomized complete block design, with three replications. The experimental field consisted of treatments of 2.3 m × 5 m plot size, with 21 cm row spacing, by Nongda 399 wheat cultivar, sown using a manual seed drill machine at a seeding rate of 225 kg ha$^{-1}$. During the second cropping season (2017–2018), wheat was grown on the same plots as the previous season, which were used after harvesting summer maize in the same field during both seasons. Basal fertilizers were applied according to local regulations with wheat special slow-release fertilizer (N–P$_2$O$_5$–K$_2$O: 24–15-5, total nutrient ≥44%) applied at 750 kg ha$^{-1}$ before sowing. Plots were irrigated thrice in both growing seasons of 2016–2017 and 2017–2018; once at tillering stage and twice at elongation stage. Weeding was continuously done by hand uniformly to all plots, while a handheld pressure sprayer was used to apply pesticides.

Solutions of Zn and/or Fe fertilizers were sprayed with different Zn/Fe ratios and doses. The spraying was done at different growth stages including booting stage (Feekes 10), anthesis (Feekes 10.51) and early filling stage (Feekes 10.54), using a handheld pressure sprayer. Spraying was done under windless conditions after sunset to avoid evaporation of the applied solution.

The main plots received different fertilizer ratios of combined or non-combined solutions of Zn used as ZnSO$_4$.7H$_2$O, and Fe used as FeSO$_4$.7H$_2$O as follows: (1) 100%Zn, (2) 80% Zn+20% Fe, (3) 60% Zn + 40% Fe, (4) 40% Zn + 60% Fe, (5) 20% Zn + 80% Fe, (6) 100% Fe. Every main plot had sub-plot treatments of fertilizer in the quantity of 0.5% (w/v) as follows: 13, 9.5, 5.5 and 0 kg ha$^{-1}$. The control plots (0 kg ha$^{-1}$ of ZnSO$_4$.7H$_2$O and/or FeSO$_4$.7H$_2$O) were treated with a corresponding quantity of deionized water. The exact quantity of Zn and Fe applied in each treatment is presented in Table 1. The cultivar used in this study is a newly developed cultivar with strong resistance to cold, drought and diseases. Therefore, these quantities were selected and applied to test its resistance to different quantities of sprayed fertilizers.

**Table 1.** Meanings of symbol used for treatments.

| Symbol | Meaning of Symbol and Exact Quantity of Zn and Fe Applied |
| --- | --- |
| 100% Zn$_{13}$ | 3.0 kg ha$^{-1}$ of Zn |
| 100% Zn$_{9.5}$ | 2.2 kg ha$^{-1}$ of Zn |
| 100% Zn$_{5.5}$ | 1.3 kg ha$^{-1}$ of Zn |
| 80% Zn + 20% Fe$_{13}$ | 2.40 kg ha$^{-1}$ of Zn +0.52 kg ha$^{-1}$ of Fe |
| 80% Zn + 20% Fe$_{9.5}$ | 1.76 kg ha$^{-1}$ of Zn +0.38 kg ha$^{-1}$ of Fe |
| 80% Zn + 20% Fe$_{5.5}$ | 1.04 kg ha$^{-1}$ of Zn +0.22 kg ha$^{-1}$ of Fe |
| 60% Zn + 40% Fe$_{13}$ | 1.80 kg ha$^{-1}$ of Zn +1.04 kg ha$^{-1}$ of Fe |
| 60% Zn + 40% Fe$_{9.5}$ | 1.32 kg ha$^{-1}$ of Zn +0.76 kg ha$^{-1}$ of Fe |
| 60% Zn + 40% Fe$_{5.5}$ | 0.78 kg ha$^{-1}$ of Zn +0.44 kg ha$^{-1}$ of Fe |
| 40% Zn + 60% Fe$_{13}$ | 1.20 kg ha$^{-1}$ of Zn +1.56 kg ha$^{-1}$ of Fe |
| 40% Zn + 60% Fe$_{9.5}$ | 0.88 kg ha$^{-1}$ of Zn +1.14 kg ha$^{-1}$ of Fe |
| 40% Zn + 60% Fe$_{5.5}$ | 0.52 kg ha$^{-1}$ of Zn +0.66 kg ha$^{-1}$ of Fe |
| 20% Zn + 80% Fe$_{13}$ | 0.60 kg ha$^{-1}$ of Zn +2.08 kg ha$^{-1}$ of Fe |
| 20% Zn + 80% Fe$_{9.5}$ | 0.44 kg ha$^{-1}$ of Zn +1.52 kg ha$^{-1}$ of Fe |
| 20% Zn + 80% Fe$_{5.5}$ | 0.26 kg ha$^{-1}$ of Zn +0.88 kg ha$^{-1}$ of Fe |
| 100% Fe$_{13}$ | 2.6 kg ha$^{-1}$ of Fe |
| 100% Fe$_{9.5}$ | 1.9 kg ha$^{-1}$ of Fe |
| 100% Fe$_{5.5}$ | 1.1 kg ha$^{-1}$ of Fe |
| Control | 0 kg ha$^{-1}$ of Zn + 0 kg ha$^{-1}$ of Fe |

The main plots received treatments of different fertilizer ratios of combined or non-combined solutions of Zn and Fe, used as ZnSO$_4$.7H$_2$O and FeSO$_4$.7H$_2$O, respectively as follows: (1) 100%Zn, (2) 80% Zn+20% Fe, (3) 60%Zn+40%Fe, (4) 40%Zn+60%Fe, (5) 20%Zn+80%Fe, (6) 100%Fe. Every main plot had sub-plot treatments that received fertilizer quantities of 0.5% (w/v) as follows: 13 kg, 9.5 kg, 5.5 kg and 0 kg ha$^{-1}$ of ZnSO$_4$.7H$_2$O and FeSO$_4$.7H$_2$O.

### 2.3. Sampling Procedures and Analysis

When grains reached physiological maturity, wheat plants were harvested in a 2 m$^2$ area (2 m × 1 m) in the center of each plot. After harvesting and threshing, cleaned grains were weighed with an electronic scale and converted to grain yield in kg ha$^{-1}$. In addition, thousand kernel weight (TKW) was determined using the same electronic scale. From the harvested samples, 15 spikes were taken from each plot to determine spike length, and the grain number per spike. After washing with deionized water and re-drying up to 13% moisture content [54], whole grains were milled into flour to determine crude fat, crude fiber, crude protein, and Zn and Fe concentration.

In the laboratory, crude protein content was determined by the Kjeldahl procedure using a K9840 Kjeldahl Analyzer (Hanon Shandong Scientific Instruments Co., Ltd., Jinan, Shandong, China) [55]. The process involved 3 major steps: First, the sample was digested in boiling concentrated H$_2$SO$_4$ at 380 °C for 90 min, with the addition of a catalyst, until complete dissolution and oxidation. The nitrogen was transformed into ammonium sulfate. Then, distillation was done by adding an excess of NaOH solution (lye). Finally, the ammonia was determined with a volumetric acid solution or by back titration. Crude protein content was obtained by multiplying total N by a factor of 5.70 [56].

Crude fiber analysis was done according to the filter bag technique [57] adapted to the F10 Crude Fiber Analyzer. It determines the organic residue remaining after digestion with 0.255 N H$_2$SO$_4$ and 0.313 N NaOH. Approximately 1 g was packed and sealed into filter bags marked with an acid-resistant marker. The bags were soaked in petroleum ether for 10 min to extract fat from the samples. Secondary bags were attached to a sample holder for 40 min of heat and agitation at 100 °C in 0.255 N H$_2$SO$_4$ and 0.313 N NaOH solutions consecutively in the vessel. After acid and base heating, the solutions were exhausted and samples were washed three times with water at the same temperature for 5 min. Then after heat and agitation in acid and base solutions, samples were are dried in an oven at 125 °C for 2 h, and ashed for 2 h at 600 ± 15 °C. Finally, the samples were weighed to calculate the crude fiber percentage:

$$\% \text{ crude fiber } = \frac{(z - (x \times 0.992)) \times 100}{y} \tag{1}$$

where $x$ is the bag weight, $y$ is the sample weight, $z$ is the loss of weight on ignition of bag/sample, and 0.992 is the blank bag ash correction.

Crude fat was analyzed using Soxhlet extraction, submerging samples in boiling solvent (petroleum ether of 30–60 °C of boiling range) that dissolves fats, oils, pigments, and other soluble substances, collectively termed "crude fat". The continuous flow of condensed solvent extracts solubilized extractables, and the resulting crude fat residue was determined gravimetrically after drying [58]. Crude fat was calculated by the following formula:

$$\% \text{ Crude fat } = \frac{F - T}{S} \times 100 \tag{2}$$

where $F$ is weight of the used cup + fat residue, $T$ is the weight of the empty cup, and $S$ is the test portion weight.

Zinc and Fe concentrations were determined using atomic absorption spectrometry (AA320CRT; Shanghai Analytical Instrument Overall Factory, Shanghai, China) [59]. Every material was soaked in nitric acid and washed with deionized water before being used, to avoid any contamination. Dry grain samples were ground with a mixer mill (MM400, Retsch, GmbH, Haan, Germany). Samples in crucibles were put over a hotplate until smoking ceased before being placed into a furnace and incinerated at 550 °C for 6 h. After cooling, the ash was dissolved in 5 ml 1:1 (v:v) HNO$_3$, and the inner wall of the container was washed, then transferred to a 50 ml volumetric flask with deionized water. Finally, the test solution was directly measured by atomic absorption spectrometry.

The instrument parameters were as follows: Wavelength, 213.9 nm and 248.3 nm for Zn and Fe, respectively; sit width, 1.3 nm for Zn and 0.2 nm for Fe; lamp current, 5 mA for Zn and 12.5 mA

for Fe; photomultiplier tube negative high voltage, 349 V for Zn and 435 V for Fe. The analysis conditions were as follows: Type of flame, air–$C_2H_2$; gas flow rate, 1.8 L min$^{-1}$; gas pressure, 160 kPa; burner height, 7.5 mm; delay time, 2 s and data collection time, 1 s. The limit of detection (LOD) and limit of quantification (LOQ) were calculated for Fe and Zn concentration by multiplying the standard deviation by 3 (LOD) and by 10 (LOQ), and dividing by the slope of the analytical curve. The concentration of samples was higher than LOD and LOQ. To evaluate the trueness and precision of the analytical method, spike tests were performed, as there was no access to certified references. The relative standard deviation values were both below 10%, and the recovery percentages were 103.4% and 99.7 % for Zn and Fe, respectively. LOD was 0.10 and 0.19 mg kg$^{-1}$ and LOQ was 0.31 and 0.58 mg kg$^{-1}$ for Zn and Fe, respectively.

## 2.4. Statistical Analysis

Data were subjected to the analysis of covariance (ANCOVA) of the effect of Zn and Fe fertilizer ratio and quantity and their interactions. Means comparisons were done by a two-sided Dunnett's test, comparing all means against the control using PASW Statistics 18 software. Graphs were made using the PASW Statistics 18 software and Excel 2016.

## 3. Results

Grain yield was not significantly affected by cropping year, nor its interaction with fertilizer ratio and fertilizer quantity, while however, fertilizer ratio and its interaction with fertilizer quantity significantly affected grain yield, TKW, spike length, crude protein, Zn content, and Fe content. That the fertilizer quantity affected spike length, crude fiber and the Zn content is shown in Table 2. Crude fiber was affected only by fertilizer quantity. The interaction of cropping year, fertilizer ratio, and fertilizer quantity did not have a significant impact on all studied variables (Table 2). The following tables separate cropping years to see the effect of fertilizer ratio and fertilizer quantity clearly. The detailed table of analysis of covariance of the effects of experimental factors and their interactions on grain yield, TKW, spike length and kernels per spike are presented in supplementary materials Table S1.

**Table 2.** Analysis of covariance (ANCOVA) of the effects of fertilizer ratio and quantity and their interactions on grain yield, thousand kernel weight (TKW), spike length, kernels per spike, crude fat, crude fiber, crude protein, Zn content and Fe in 2 cropping seasons.

| Source of Variation | Grain Yield | TKW | Spike Length | Kernels per Spike | Crude Fat | Crude Fiber | Crude Protein | Zn Content | Fe Content |
|---|---|---|---|---|---|---|---|---|---|
| Year (Y) | ns | ns | ns | ns | ns | ns | ns | ns | ns |
| Fertilizer ratio (R) | *** | *** | *** | ns | ns | ns | *** | *** | *** |
| Fertilizer quantity (Q) | ns | ns | ** | ns | ns | *** | ns | *** | ns |
| Y × R | ns | ns | ns | ns | ns | ns | ns | ns | ns |
| Y × Q | ns | ns | ns | ns | ns | ns | ns | ns | ns |
| R × Q | * | * | ** | ns | ns | ns | * | *** | *** |
| Y × R × Q | ns | ns | ns | ns | ns | ns | ns | ns | ns |

ns: non-significant, * significant at $p < 0.05$, ** significant at $p < 0.01$, *** significant at $p < 0.001$.

Compared with the control, during the first cropping season (2016–2017), 100% Fe$_{13}$ treatment influenced grain yield, while in the second cropping season (2017–2018), different Zn and Fe treatments, either mixed or not, affected the yield. In addition, treatments containing high Fe content significantly increased the yield, i.e., solutions with Fe content ≥60%, especially 100% Fe$_{13}$, followed by 20% Zn + 80% Fe$_{5.5}$ with 61.59% and 39.19% increases, respectively, in the first cropping year. In the second cropping season, 100% Fe$_{13}$ still outperformed, followed by 40% Zn + 60% Fe$_{5.5}$ with 63.90% and 41.21% increases, respectively (Table 3).

Averaged across the two cropping seasons, 100%Fe$_{13}$ outweighs, followed by 20% Zn + 80% Fe$_{5.5}$, while the lowest yield was observed with 80% Zn + 20%Fe$_{9.5}$, with a significant negative effect on yield (Figure 2). One can observe that Fe tends to increase yield more than Zn does. Generally, foliar application of Zn and Fe fertilizers showed significant effects in the second year over the first year. It was observed that yield was increased proportionally with the quantity of 100% Fe fertilizer as follows: 100% Fe$_{13}$ > 100% Fe$_{9.5}$ > 100% Fe$_{5.5}$. Ratio 80% Zn + 20% Fe and 60% Zn + 40% Fe presented relatively the same effects, which had no statistically significant effect over the control. Ratio 40% Zn + 60% Fe and 20% Zn + 80% Fe showed almost the same influence on the yield, and their effect was better than that of 80% Zn + 20% Fe and 60% Zn + 40% Fe. According to quantities, 5.5 kg presented higher yield, while 9.5 kg had lower yield in both ratios.

**Table 3.** Grain yield, thousand kernel weight (TKW), spike length and kernels per spike means comparison against control, as affected by foliar application of Zn and Fe fertilizers [a].

| Treatments [b] | 2016–2017 | | | | 2017–2018 | | | |
|---|---|---|---|---|---|---|---|---|
| | Grain Yield (t ha$^{-1}$) | TKW (g) | Spike Length (cm) | Kernels per Spike | Grain Yield (t ha$^{-1}$) | TKW (g) | Spike Length (cm) | Kernels per Spike |
| 100% Zn$_{13}$ | 4.228 | 44.07 | 7.88 | 35.20 | 4.106 *** | 44.20 * | 7.91 *** | 35.21 |
| 100% Zn$_{9.5}$ | 3.829 | 43.96 | 7.79 | 38.85 | 3.800 *** | 44.08 | 7.83 *** | 38.38 *** |
| 100% Zn$_{5.5}$ | 3.711 | 46.15* | 8.35 | 38.85 | 3.590 ** | 46.01 *** | 8.32 | 38.72 *** |
| 80% Zn + 20% Fe$_{13}$ | 3.395 | 43.39 | 8.19 | 36.25 | 3.347 | 43.18 | 8.21 | 36.24 |
| 80% Zn + 20% Fe$_{9.5}$ | 2.785 | 43.62 | 8.43 | 36.75 | 2.682 * | 43.51 | 8.44 ** | 36.62 * |
| 80% Zn + 20% Fe$_{5.5}$ | 3.332 | 44.25 | 8.49 | 40.10 | 3.361 | 44.08 | 8.43 ** | 39.84 *** |
| 60%Zn + 40% Fe$_{13}$ | 2.919 | 43.01 | 8.04 | 35.55 | 2.869 | 42.74 | 7.98 ** | 34.99 |
| 60% Zn + 40% Fe$_{9.5}$ | 2.905 | 42.68 | 7.86 | 34.25 | 3.007 | 42.71 | 7.94 ** | 34.63 |
| 60% Zn + 40% Fe$_{5.5}$ | 3.157 | 43.28 | 8.03 | 34.85 | 3.064 | 43.32 | 7.92 *** | 34.21 |
| 40% Zn + 60% Fe$_{13}$ | 4.051 | 41.68 | 8.01 | 38.00 | 3.955 *** | 41.46 ** | 7.94 ** | 37.78 *** |
| 40% Zn + 60% Fe$_{9.5}$ | 3.604 | 42.26 | 7.99 | 36.75 | 3.619 *** | 42.38 | 8.05 | 36.70 * |
| 40% Zn + 60% Fe$_{5.5}$ | 4.161 | 42.63 | 8.13 | 34.75 | 4.320 *** | 42.72 | 8.09 | 35.23 |
| 20% Zn + 80% Fe$_{13}$ | 4.007 | 40.54 | 8.68 | 37.50 | 4.159 *** | 40.49 *** | 8.72 *** | 37.12 * |
| 20% Zn + 80% Fe$_{9.5}$ | 3.685 | 41.81 | 8.65 | 36.05 | 3.805 *** | 41.80 ** | 8.63 *** | 36.49 * |
| 20% Zn + 80% Fe$_{5.5}$ | 4.251 | 43.76 | 8.39 | 39.85 | 4.316 *** | 43.55 | 8.38 * | 39.29 *** |
| 100% Fe$_{13}$ | 4.935 * | 43.61 | 8.18 | 36.85 | 5.050 *** | 43.76 | 8.09 | 37.13 ** |
| 100% Fe$_{9.5}$ | 4.228 | 41.21 | 8.55 | 36.40 | 4.037 *** | 41.03 *** | 8.56 *** | 35.94 |
| 100% Fe$_{5.5}$ | 3.941 | 44.16 | 8.22 | 37.65 | 4.022 *** | 44.26 * | 8.17 | 37.28 *** |
| Control | 3.054 | 43.07 | 8.19 | 35.67 | 3.081 | 43.02 | 8.19 | 35.15 |

[a]: Values are means of three replicates. [b]: Combination of Zn and Fe fertilizers described in Table 1. Sprayed three times, * significant at $p < 0.05$, ** significant at $p < 0.01$, *** significant at $p < 0.001$

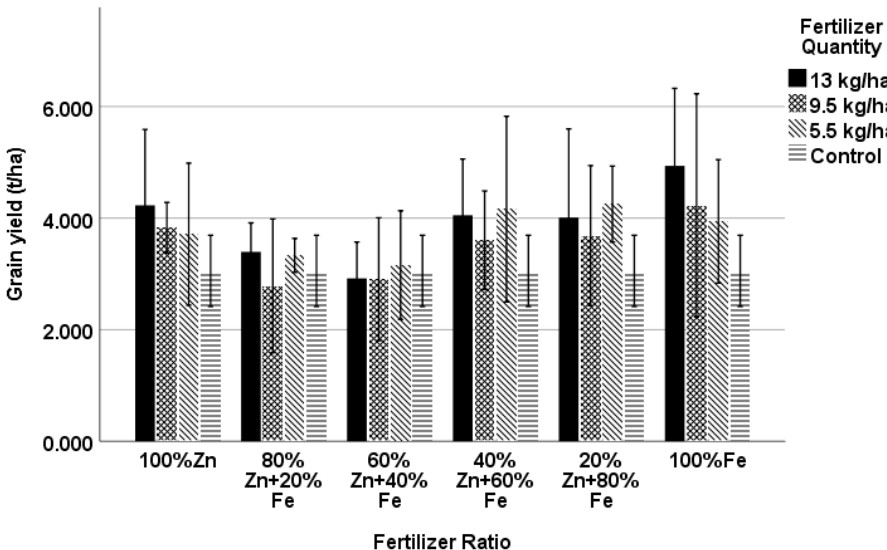

**Figure 2.** Effect of combined foliar application of Zn and Fe on yield of winter wheat (Nongda 399 cultivar). Values are means of two cropping seasons. Vertical bars represent the 95% confidence interval.

Largely, the data revealed that TKW was not affected by foliar application in the first year, with the exception of 100% $Zn_{5.5}$, which increased TKW significantly, whereas, in the second year, the treatments showed better performance, with 100% $Zn_{13}$, 100% $Zn_{5.5}$, and 100% $Fe_{5.5}$ increasing TKW by 2%, 7% and 3%, respectively. Other treatments i.e., 40% Zn + 60% $Fe_{13}$, 20% Zn + 80% $Fe_{13}$, 20% Zn + 80% $Fe_{9.5}$, and 100% $Fe_{9.5}$, significantly decreased TKW by 3%, 5%, 2%, and 4%, respectively (Table 3).

Compared to other fertilizer ratios, 100% Zn and 60% Zn + 40% Fe decreased spike length significantly, while 20% Zn + 80% Fe and 100% Fe increased spike length significantly in most fertilizer quantities in the second season (Table 3). The number of kernels per spike was not considerably affected in the first year, but was significantly increased in the second year by a number of treatments: Some of them were significant at 0.001, 0.01 and 0.05. Furthermore, 80% Zn + 20% $Fe_{5.5}$ showed a 13% increase followed by 20% Zn + 80% $Fe_{5.5}$ which increased by 11%. It is important to note that all levels of 60% Zn + 40% Fe did not significantly affect the number of kernels per spike (Table 3).

Throughout the two years, the foliar application of Zn and Fe fertilizers did not considerably affect crude fat for all treatments (Table 4). The average means of treatments of those two years are plotted in Figure 3a.

Crude fiber content was significantly increased by a foliar application of Zn and Fe fertilizers in all treatments, but most remarkably up to three-fold with the 60% Zn + 40% $Fe_{5.5}$ treatment. Moreover, 80% Zn + 20% Fe and 60% Zn + 40% Fe fertilizer ratios have outweighed other ratios; even though most ratios were statistically significant, but 80% Zn + 20% Fe and 60% Zn + 40% Fe values were higher (Figure 3c). In addition, some treatments were not statistically significantly different from the control: 100% $Zn_{9.5}$, 40% Zn + 60% $Fe_{5.5}$, and 20% Zn + 80% $Fe_{13}$ in both years. The lowest value was observed in the control treatment (Table 4).

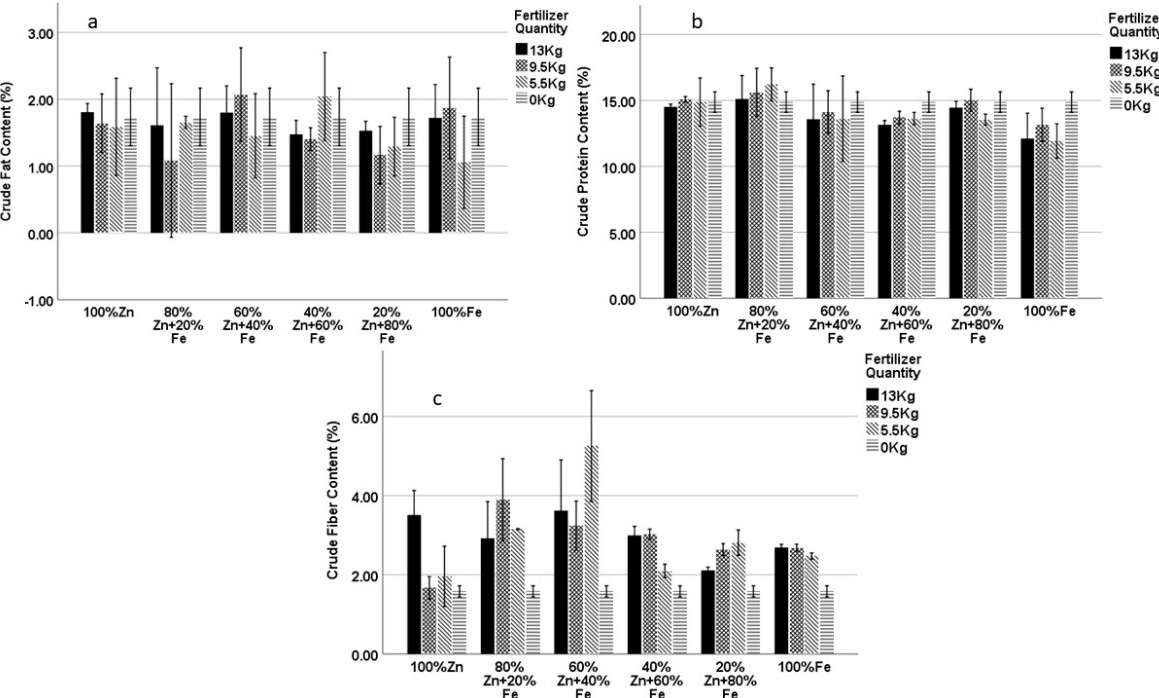

**Figure 3.** Crude fat (**a**), crude protein (**b**), and crude fiber (**c**) content in wheat grains as affected by foliar application of combined Zn and Fe fertilizers. Plotted values are means of two cropping seasons. Vertical bars represent the 95% confidence interval.

**Table 4.** Means comparison of crude fat, crude fiber, and crude protein of whole grain winter wheat affected by foliar application of Zn and Fe fertilizers [a].

| Treatments [b] | 2016–2017 | | | 2017–2018 | | |
|---|---|---|---|---|---|---|
| | Crude Fat (%) | Crude Fiber (%) | Crude Protein (%) | Crude Fat (%) | Crude Fiber (%) | Crude Protein (%) |
| 100% $Zn_{13}$ | 1.80 | 3.51 *** | 14.51 | 1.80 | 3.58 *** | 14.49 |
| 100% $Zn_{9.5}$ | 1.64 | 1.67 | 15.08 | 1.88 | 1.77 | 15.09 |
| 100% $Zn_{5.5}$ | 1.58 | 1.96 *** | 14.87 | 1.78 | 2.17 | 14.99 |
| 80% Zn + 20% $Fe_{13}$ | 1.61 | 2.92 *** | 15.11 | 1.71 | 2.96 *** | 15.00 |
| 80% Zn + 20% $Fe_{9.5}$ | 1.08 | 3.89 *** | 15.60 | 1.46 | 4.13 *** | 15.49 * |
| 80% Zn + 20% $Fe_{5.5}$ | 1.65 | 3.15 *** | 16.21 | 1.95 | 3.29 *** | 16.28 *** |
| 60%Zn + 40% $Fe_{13}$ | 1.80 | 3.62 *** | 13.57 | 2.07 | 3.63 *** | 13.40 ** |
| 60% Zn + 40% $Fe_{9.5}$ | 2.07 | 3.24 *** | 14.12 | 2.02 | 3.31 *** | 14.21 |
| 60% Zn + 40% $Fe_{5.5}$ | 1.45 | 5.25 *** | 13.62 | 1.86 | 5.53 *** | 13.41 ** |
| 40% Zn + 60% $Fe_{13}$ | 1.47 | 2.99 *** | 13.15 | 1.57 | 3.15 *** | 13.17 *** |
| 40% Zn + 60% $Fe_{9.5}$ | 1.40 | 3.02 *** | 13.70 | 1.93 | 3.17 *** | 13.72 * |
| 40% Zn + 60% $Fe_{5.5}$ | 2.04 | 2.09 | 13.62 | 2.37 | 2.25 | 13.59 * |
| 20% Zn + 80% $Fe_{13}$ | 1.53 | 2.11 | 14.45 | 1.86 | 2.26 | 14.42 |
| 20% Zn + 80% $Fe_{9.5}$ | 1.16 | 2.63 ** | 15.01 | 1.35 | 2.75 *** | 15.05 |
| 20% Zn + 80% $Fe_{5.5}$ | 1.29 | 2.81 *** | 13.54 | 1.75 | 2.91 *** | 13.51 ** |
| 100% $Fe_{13}$ | 1.72 | 2.69 ** | 12.10 ** | 1.90 | 2.82 *** | 11.98 *** |
| 100% $Fe_{9.5}$ | 1.87 | 2.68 ** | 13.15 | 1.98 | 2.81 *** | 13.07 *** |
| 100% $Fe_{5.5}$ | 1.05 | 2.47 * | 11.92 *** | 1.16 | 2.61 ** | 11.84 *** |
| Control | 1.73 | 1.58 | 14.87 | 2.14 | 1.70 | 14.57 |

[a]: Values are means of three replicates. [b]: Combination of Zn and Fe fertilizers described in Table 1. Sprayed three times, * significant at $p < 0.05$, *** significant at $p < 0.01$, ***significant at $p < 0.001$.

Results show that crude protein was significantly improved by foliar application of Zn and Fe fertilizers only in the second year in the 80% Zn + 20% $Fe_{9.5}$ and 80% Zn + 20% $Fe_{5.5}$ treatment by 6% and 12%, respectively. Specifically, 80% Zn + 20% $Fe_{5.5}$ was the outstanding treatment in increasing crude protein. Treatments consisting of Zn only 100% $Zn_{13}$, 100% $Zn_{9.5}$ and 100% $Zn_{5.5}$ did not considerably affect the crude protein content of whole grain. Treatments containing Fe only 100% $Fe_{13}$, 100% $Fe_{9.5}$, 100% $Fe_{5.5}$ significantly reduced crude protein content up to 13% with 100% $Fe_{5.5}$, and this value was the lowest among treatments. Treatments of 60% Zn + 40 % Fe, 40% Zn + 60% Fe and 20% Zn + 80% Fe fertilizer ratios, either did not significantly increase crude protein content or rather it decreased crude protein content (Table 4). Fertilizer ratio 60% Zn + 40% Fe and 40% Zn + 60% Fe in their respective quantities showed almost the same effect upon crude protein, which is better than that of 100% Fe, but not as good as that of 100% Zn, 80% Zn + 20% Fe and 20%Zn + 80%Fe. Averaged across these two years, the plotted means shows that treatments of the 100% Fe ratio had the lowest crude protein content, while those of the 80% Zn + 20% Fe ratio presented the highest values (Figure 3b). The table of analysis of covariance of the effects of experimental factors and their interactions on crude fat, crude fiber and crude protein are presented in Table S2 of supplementary materials.

The concentration of Zn in grain was significantly influenced by foliar application of Zn and Fe fertilizers in both years (Table 5). The average of grain Zn concentration increased from 19.79 mg $kg^{-1}$ in the control to 38.79 mg $kg^{-1}$ in the 100% $Zn_{9.5}$ treatment during the first year, and from 19.25 mg $kg^{-1}$ in the control to 39.63 mg $kg^{-1}$ in the 100% $Zn_{9.5}$ treatment in the second year, which is an increment of 96.9% and 105.8%, respectively. The highest increase was observed in treatments with only Zn or a high Zn fertilizer ratio particularly in 100% $Zn_{9.5}$, and the lowest increase was found in treatments with low or no Zn. The lowest Zn concentration was observed in the 100% $Fe_{13}$ treatment during the two years (Table 5). Zinc mean values of the two seasons show that when Zn concentration in sprayed solution decreased, it tended to decrease the grain Zn concentration. Hence, the 100% Fe ratio shows the smallest values of Zn concentration compared to other ratios, while 100% Zn shows higher Zn concentration values (Figure 4).

**Table 5.** Means comparison of Zn and Fe content of whole grain winter wheat affected by foliar application of Zn and Fe fertilizers [a].

| Treatments [b] | 2016–2017 | | 2017–2018 | |
|---|---|---|---|---|
| | Zn (mg kg$^{-1}$) [c] | Fe (mg kg$^{-1}$) | Zn (mg kg$^{-1}$) | Fe (mg kg$^{-1}$) |
| 100% Zn$_{13}$ | 37.74 *** | 36.34 | 37.46 *** | 35.91 |
| 100% Zn$_{9.5}$ | 38.79 *** | 43.22 | 39.63 *** | 43.02 *** |
| 100% Zn$_{5.5}$ | 35.12 *** | 49.26 ** | 34.17 *** | 48.32 *** |
| 80% Zn + 20% Fe$_{13}$ | 25.75 | 41.52 | 27.08 *** | 40.90 * |
| 80% Zn + 20% Fe$_{9.5}$ | 29.97 * | 46.32 * | 30.64 *** | 46.87 *** |
| 80% Zn + 20% Fe$_{5.5}$ | 26.04 | 44.14 | 26.12 *** | 45.00 *** |
| 60%Zn + 40% Fe$_{13}$ | 32.52 ** | 42.27 | 32.51 *** | 41.40 ** |
| 60% Zn + 40% Fe$_{9.5}$ | 27.90 | 42.30 | 26.72 *** | 41.45 *** |
| 60% Zn + 40% Fe$_{5.5}$ | 26.12 | 43.83 | 25.49 *** | 42.27 *** |
| 40% Zn + 60% Fe$_{13}$ | 26.54 | 43.10 | 26.74 *** | 43.27 *** |
| 40% Zn + 60% Fe$_{9.5}$ | 25.78 | 42.83 | 25.44 *** | 42.47 *** |
| 40% Zn + 60% Fe$_{5.5}$ | 23.66 | 42.27 | 22.72 ** | 42.10 *** |
| 20% Zn + 80% Fe$_{13}$ | 22.70 | 42.51 | 22.41 * | 42.41 *** |
| 20% Zn + 80% Fe$_{9.5}$ | 26.66 | 45.31 | 26.27 *** | 45.40 *** |
| 20% Zn + 80% Fe$_{5.5}$ | 25.27 | 47.17 * | 25.04 *** | 47.33 *** |
| 100% Fe$_{13}$ | 17.41 | 46.50 * | 17.10 | 46.15 *** |
| 100% Fe$_{9.5}$ | 17.99 | 44.05 | 17.49 | 44.20 *** |
| 100% Fe$_{5.5}$ | 26.84 | 43.28 | 25.15 *** | 42.16 *** |
| Control | 19.79 | 38.41 | 19.25 | 38.18 |

[a]: Values are means of three replicates plots. [b]: Zn and Fe fertilizers applied three times; [c]: Zn and Fe contents values are expressed in mg per kg of dry weight; * significant at $p < 0.05$, ** significant at $p < 0.01$, *** significant at $p < 0.001$.

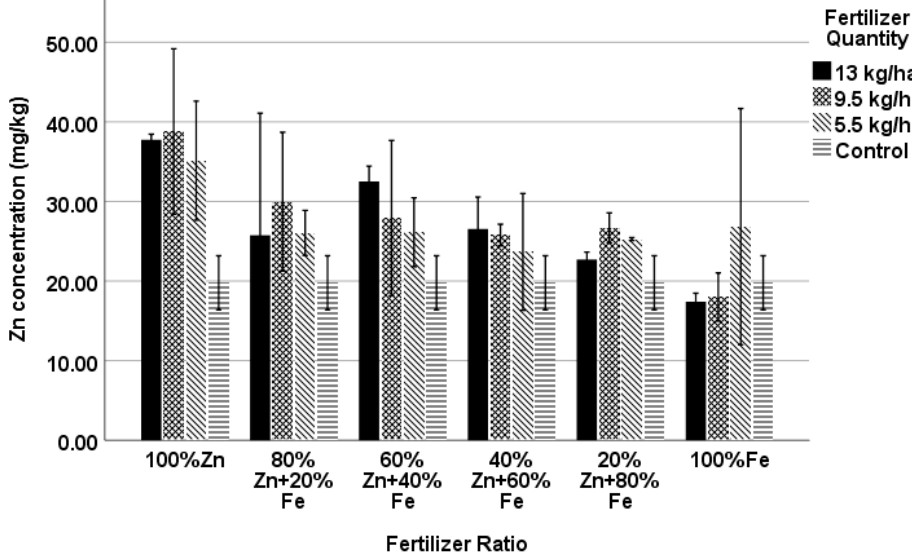

**Figure 4.** Effect of foliar application of combined Zn and Fe fertilizers on wheat grain Zn concentration. Values plotted are means of two copping seasons. Vertical bars represent the 95% confidence interval.

The concentration of Fe was significantly influenced by a foliar application of combined Zn and Fe fertilizers in both years, but mostly in the second year (Table 5). Average concentration of treatments compared to control in the first season shows that Fe was significantly increased in 100% Zn$_{5.5}$, 80% Zn + 20% Fe$_{9.5}$, 20% Zn + 80% Fe$_{5.5}$ and 100% Fe$_{13}$, and the increase varied between 8 and 28%, according to treatment, with the exception of 100% Zn$_{13}$, which decreased Fe concentration by 5% (Table 5). Average Fe concentration in the second season improved from 38.18 mg kg$^{-1}$ in control to 48.32 mg kg$^{-1}$ in 100% Zn$_{5.5}$, and its increase varies between 8.4 and 26.5% across treatments. Generally, Fe concentration was not improved by the combination of Zn and Fe in the first year, but an increase was

observed in the second year. Fe applied alone significantly improved with 100% $Fe_{5.5}$ only in the first year, and in the second year, all quantities did significantly improve grain Fe concentration (Table 5). During the two seasons, 60% Zn + 40% Fe and 40% Zn + 60% Fe ratios seemed to have almost the same effect on Fe grain concentration, which was a small increase compared to other ratios (Figure 5). The analysis of covariance of the effects of experimental factors and their interactions on Zn and Fe concentration in grain can be found in Table S2 of supplementary materials.

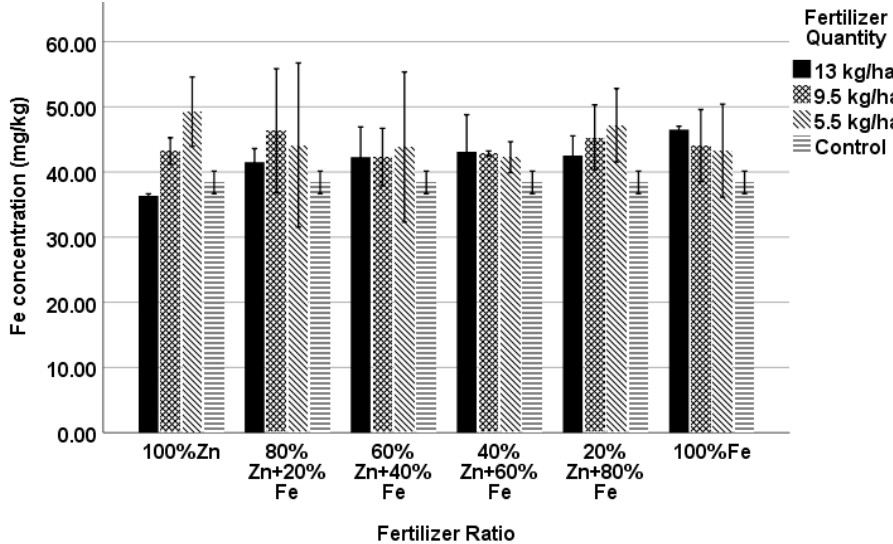

**Figure 5.** Concerning the foliar application of combined Zn and Fe fertilizers on wheat grain Fe concentration. Values plotted are the means of both copping seasons. Vertical bars represent the 95% confidence interval.

## 4. Discussion

### 4.1. Yield and Yield Components

In this study, grain yield was significantly improved by some Zn and Fe foliar fertilizer treatments. Not much difference was observed in the first season, but in the second season there was reasonable significance especially with 20% Zn + 80% Fe and 100% Fe in their respective quantities. This could be explained by the fact that the plant response to fertilization or micronutrient deficiency is greatly affected by seasonal changes in climatic conditions [60].

Compared to the control, 100% Zn treatments significantly increased the grain yield in the second cropping season, even though the increase was not significant in the first cropping year. Zn was previously reported to increase grain yield [61]. The first year results correlate with those reported by Zhang et al. [62] and Li et al. [50], who argued that yield was unaffected by any application of Zn on both medium and high Zn and high Fe soil concentrations in China. Cakmak et al. [27] in Turkey also reported that in Samsun and Adana (medium to high DTPA-Zn soil concentration), the grain yield was unaffected by foliar application of Zn, while in Konya (low DTPA-Zn soil concentration), the grain yield was increased by 23% and 21% in two different years. The inability to affect the yield and yield components might be due to relatively high DTPA-extractable Zn and Fe concentrations in the experimental soils. In contrast, despite the high DTPA-Zn and Fe soil concentration, our study in the second years presented an increase in grain yield.

In addition, yield components resulted in biochemical reactions and a relocation of photosynthetic products [63]. It is believed that through photosynthesis and sugar transformations, Zn exerts an effect on carbohydrates metabolism [64], and Zn deficiency may result in reductions in net photosynthesis by 50–70%, depending on the plant species and the severity of the deficiency, inducing dramatic changes in yield components [64]. Zinc in plants plays a major role as a functional, structural or

regulatory co-factor of many of enzymes, and is involved in sucrose and starch formation [64,65]. High Fe concentrations (100% Fe) significantly increased yield, and the best yield was observed with 100% Fe[13]. All of the 20% Zn + 80% Fe and 40% Zn + 60% Fe treatments were also significant in increasing the yield, even though the difference over the control was more observed in the second season.

The combination of Zn and Fe positively affected the yield in some ratios. During this study, the yield was reduced with 80% Zn + 20% Fe and 60% Zn + 40% Fe ratios, compared to the control and treatments where Zn, Fe were not combined. It was observed that 40% Zn + 60% Fe and 20% Zn + 80% Fe ratios increased the yield, which is a different effect. Iron percentage up to 40% in the solution reduced the yield, but from 60%, the yield increased. The increase of Fe up to 40% in the solution could have reduced or inhibited the activity and translocation of Zn in contributing to yield, but a higher amount of Fe actually increased grain yield [66]. The antagonism between Fe and Zn might be the reason for the yield decrease. Iron applied separately increased yield, and a high amount gave a higher yield. Partially, similar results were reported by Verma and Tripathi [67], who illustrated the decrease of grain and straw yield in the paddy; when Zn was not applied, Fe application decreased grain yield and straw, but with an addition of Zn, Fe did not affect grain and straw. The present study showed that low a percentage of Fe in the solution reduced the yield, while higher percentages increased the yield. In addition, Fe metabolism was reported to be affected by S availability in Strategy I and Strategy II plants. For instance, Zuchi et al. argued that the uptake and translocation of Fe to the shoot might be prevented by an inadequate supply of S in tomato, and the Fe deficiency led to a strong decrease in total S content in the shoots and roots of tomato [43]. Moreover, Ciaffi et al. demonstrated that augmenting the S uptake capacity of roots can maintain phytosiderophores production under Fe deficiency, and Fe deprivation resulted in a higher accumulation of S in the shoots of wheat [42]. The interaction among plant nutrients can be either synergistic, antagonistic, zero-interactive or Liebig-synergistic. These interactions illustrate that the supply of one nutrient can affect the function of another nutrient [68]. Therefore, these interactions can affect plant growth and yield [69,70].

### 4.2. Crude Fat

Crude fat was not significantly affected by the foliar application of Zn and Fe fertilizers in both cropping years. Similar results were reported by Bressani [71] when NPK 12-24-12 was applied to three amaranth species: *A. eruentus* (Rodale 82S-1034), *A. hypochondriacus* (Rodale 81S-1024), and *A. caudatus (CAC-20002)*, even though these values were of a higher range than what is usually reported.

### 4.3. Crude Fiber

Grain fiber consists of an edible plant cell wall, carbohydrates and polymers that occur naturally in the food as consumed. The best sources of fiber are whole grains and bran, especially the outer part of the grain called kernel layers, which consist of pericarp, testa and aleurone [72,73]. Crude fiber was increased remarkably by combined solution with a 60% Zn + 40% Fe ratio as well as other ratios. This could be possibly linked with a significant accumulation of Zn and Fe in aleurone, endosperm, and embryo [22,27] when these micronutrients are supplied to wheat, as discussed in different literature [13,22,74]. It can also be a result of assimilates remobilization in which process Zn and Fe have a greater role.

### 4.4. Crude Protein

The effect of the foliar application of combined Zn and Fe fertilizers on crude protein kept changing among treatments. A 100% Zn solution did not differ significantly from the control, but a solution with 80% Zn and 20% Fe ratio increased crude protein significantly. More precisely, as the fertilizer quantity increases, crude protein tends to decrease in this particular solution, which makes 80% Zn + 20% Fe[5.5] the best performer for crude protein. This enhancement in grains quality may be due to the role of microelements in maintaining balanced plant physiological growth and activation of plant enzymes. There is a fluctuating effect in solutions made with 60% Zn + 40% Fe, 40% Zn + 60% Fe,

and 20% Zn + 80% Fe fertilizer ratios; some show a significant decrease, and others a non-significant effect, which makes these solutions unsuitable for protein increment in whole grain. Moreover, the 100% Fe solution is completely not ideal, because it significantly reduced crude protein. The lowest value was found with 100% Fe$_{5.5}$. These results differ from those reported by Melash et al. [75] in a study conducted in Ethiopia which revealed that FeSO$_4$ tended to improve grain protein and gluten content as well as Zeleny index, compared to ZnSO$_4$. Micronutrients and their interactions affect the physiological processes of plants, which has a significant impact on grain yield and quality [76]. These results partially reflect what was reported by Singh et al. [8] in a pot experiment conducted in Turkey. Furthermore, Zhang et al. [77] stated that foliar Zn application has a small effect on protein concentration and gluten characteristics. Average of two cropping seasons, 60% Zn + 40% Fe, 40% Zn + 60% Fe and 20% Zn + 80% Fe and 100% Fe ratios present lower protein content than control. However, literature shows evidence that the quantity of grain protein hugely affects the grain capacity in storing Zn and Fe, which was confirmed by positive correlations between Zn, Fe and protein in different wheat genotypes [28,78,79]. This was not confirmed in the present research, as high quantities did not correspond to high protein concentration.

### 4.5. Zn Concentration

Zinc applied alone enhanced the Zn concentration of whole grain. As the concentration of Zn decreases and Fe increases in the applied solution, there is a tendency of decreasing Zn concentration in grains. Studies reported that generally, the Zn concentrations in grain within wheat-cultivated regions range from 20–35 mg kg$^{-1}$, with an average value around 28–30 mg kg$^{-1}$ [4,18]. Although the combination of Zn and Fe in different ratios and quantities relatively increased Zn concentration, treatments consisting of Zn alone registered high grain Zn concentration. The results are in agreement with Zhang et al. [77], who reported that foliar application of only 0.4% ZnSO$_4$.7H$_2$O increased Zn content in grain by 58%, and Zou et al. [48], who reported an increase of up to 83% of Zn in grain. Moreover, studies have also mentioned an increase of 27% Zn in rice and 9% in maize [18] when Zn is applied as foliar fertilization.

A possible reason for this increment is that foliar-applied Zn is phloem-mobile, and can be readily translocated into emerging grains in wheat [33,80]. The concentration of Zn decreased with high Fe quantities, and 100% Fe$_{13}$ showed the lowest Zn concentration, which could be explained by the antagonism of Fe and Zn when Fe is applied in high quantities as it was reported by Alam [81] and Alloway [64]. The same results were obtained by Ghamesi [82] in a study on Iron interaction with Copper, Zinc, and Manganese in wheat. Similarly, Verma and Tripathi [67] reported the same observations for a submerged paddy, as well as Brar and Sheklon [66], who reported that Zn translocation decreased as Fe increased. Aciksoz et al. [78] reported that in their study, Fe spray caused a not large decrease in grain Zn concentration, which was described as an undesirable effect of foliar Fe spray on grain Zn. Moreover, it was reported that foliar application of Fe decreased the concentration of Zn in the leaves and roots of tomato [83]. In the present study, 100% Fe$_{13}$ and 100% Fe$_{9.5}$ registered lower Zn concentration than control, which highlights the negative effect of Fe foliar application on grain Zn.

### 4.6. Fe Concentration

Generally, the foliar application of combined Zn and Fe fertilizers improved grain Fe concentration in both years. However, only 100% Zn$_{5.5}$, 80% Zn + 20% Fe$_{9.5}$, 20% Zn + 80% Fe$_{5.5}$ and 100% Fe$_{13}$ improved Fe concentration in the first year, while almost all except treatments 100% Zn$_{13}$ enhanced grain Fe concentration in the second year. An increase of Fe by Zn application was also reported by Wang et al. [76], where the foliar application of Zn resulted in a significant increase in Fe concentration. Despite a strong correlation between grain Zn and Fe, recently Cakmak et al. [22] reported that, in contrast to the positive effects of Zn fertilization on grain Zn concentrations, soil or foliar applications of various organic and inorganic forms of Fe fertilizers cannot influence grain Fe concentrations of durum wheat. Our results reflected partially the same observations in the first year, but the fact was

not verified in the second year. The effect of Fe in increasing grains Fe concentration is still debatable. While in China [54] and Iran [84] foliar spray of $FeSO_4$ increased grain Fe concentration by 21% and 14%, respectively, it showed no effect on grain Fe concentration in Canada [85]. Moreover, Aciksoz et al. believe that, in their study, the increase in grain Fe concentration of 14% and 10% from FeEDTA and $FeSO_4$ respectively, was due to the plant N status [78]. The high increase of grain Fe concentration was observed in a 100% Zn ratio, but it was inversely proportional to its quantities; 100% $Zn_{5.5}$ > 100% $Zn_{9.5}$ > 100% $Zn_{13}$. This gives an idea that high quantities of Zn spray fertilizers are not ideal in increasing the grain Fe. The increase with 100% Zn treatments could be explained by the simultaneous increase of Fe, due to a translocation of Zn from vegetative parts to grains, which is coordinated by same transporters, even though this Fe might not contribute a lot to flour Fe concentration, due to its peripheral location in grain [54,86]. Most of the times it is removed during the flour making process. Same observations were made by Li et al. [50] in the study conducted in China for assessing Zn and Fe concentrations in wheat flour affected by foliar applications of Zn, and macronutrients and the same mechanism might have contributed to other ratio as well.

## 5. Conclusions

In the present study, 19 treatments were used including the control, Zn, Fe and combined Zn and Fe. Different combinations of fertilizer ratios and quantities were studied, and a possible increase was observed in these study variables. It was observed that Zn, combined with small percentage of Fe in the solution, reduced the yield, while a higher percentage increased the yield. Zn applied alone was better in increasing Zn concentration in wheat grains, and higher quantities tended to highly increase its concentrations in grains. Fe applied alone has a good effect on its concentration in grains, but it did not differ much in the effects of treatments containing a low quantity of Zn. Combined Zn and Fe treatments increased Zn and Fe content, but applying these micronutrients separately was still better than combining them to increase their concentration in wheat grain. The consistent positive impact was also seen in crude protein, and crude fiber, while crude fat was unaffected by any treatment.

However, there was a fluctuation among results with the same fertilizer ratio, and some observations do not reflect what was expected, such as a correlation between Zn and protein. Therefore, further studies are required to explore the genetic implications of this combination of fertilizers upon grain quality.

**Supplementary Materials:** The following are available online at http://www.mdpi.com/2073-4395/9/5/250/s1, Table S1: Analysis of covariance of the effects of experimental factors and their interactions on grain yield, Thousand Kernel Weight (TKW), spike length and kernels per spike, Table S2: Analysis of covariance of the effects of experimental factors and their interactions on crude fat, crude fiber, crude protein, Zinc content and iron content.

**Author Contributions:** Conceptualization, E.N. and J.B.H.; Formal analysis, E.N. and A.T.; Investigation, E.N., A.T., I.M., W.A.N., Y.P.Y. and B.M.W.; Methodology, E.N. and J.B.H.; Project administration, J.B.H.; Writing—original draft, E.N.; Writing—review & editing, E.N.

**Funding:** This research was funded by the National high technology development plan (863) (2013AA102902).

**Conflicts of Interest:** The authors declare no conflict of interest.

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
