# Peer review of "Winter Wheat Grain Quality, Zinc and Iron Concentration Affected by a Combined Foliar Spray of Zinc and Iron Fertilizers"

_agronomy, doi:10.3390/agronomy9050250_

Reviewer 1 Report

see file attached

Author Response

Response to Reviewer 1 Comments (Minor revision)

Manuscript title: Winter Wheat Grain Quality, Zinc and Iron Concentration Affected by a Combined Foliar Spray of Zinc and Iron Fertilizers

Response to comments:

All comments mentioned in the PDF file provided by reviewer were addressed. Given that most of them were minor errors, they were corrected and highlighted in the revised version of the manuscript and can be easily traced by using Trace change tool under Review menu of the MS word.

However, there are some points that needs more clarifications:

Point 1. Page 5: add 13, 9.5, 5.5 to Zn in the tables for names of treatments.

Response 1: The reviewer proposed to add the quantity of fertilizer used per ha in the symbol used for treatments names. We were not able to apply this to the names because it might look as if 13 kg ha-1(any used quantity) of Zn was mixed to 13 kg ha-1(any used quantity) of Fe and the % loses its meaning.

80% Zn + 20% Fe13: this is a single name for a treatment made of 13kg ha-1 made of 80% ZnSO4.7H2O + 20% FeSO4.7H2O

Point 2. Page 9: why does the ordinate of figure 3i start from -1

Response 2: This is because the is a high variability of value of crude fat content and this variability range affect the confidence interval. It does not mean that any value of crude fat content is under 0%.

Point 3. Page 9: Unclear. What does the increase of 14.7 and 16.4 refer to?

Response 3: after crosschecking the results, I find this was insertion of irrelevant data as there is no range of increment to only one comparison mentioned between control and another treatment. Therefore, these numbers were dropped and only increment % were kept.

 Point 4. Page 12: The soil used is classified as high Zn and Fe content. This point is crucial. Explaining in more detail why it was obtained an increase of yield with the foliar fertilization with Zn and Fe.

Response 4: More references were given supporting the results where grain yield was unaffected by foliar application of Zn and Fe in medium to high Zn soils. The second year of our study behaved differently and the cause of this behavior remain unknown to us as the second year had same cultivation practices as the first one.

Reviewer 2 Report

Point 1: The Introduction does not include relevant references regarding the topic of "agronomic biofortication technologies". In particular, not only nitrogen but also sulfur is important to facilitate the uptake of Fe and Zn. To this purpose, I suggest you to mention two papers: (1) Celletti et al., 2016 and (2) Astolfi et al., 2018, in which you can read about Fe accumulation in graminaceous plants and in their grains, that is affected by adequate and excess sulfur supply.

Point 2: In the first line of Introduction, replace"produced" with "cultivated", please

Point 3: I recommend you to change some words that you were wrong to write in the Introduction:

"Foot" with "food";

"en masse" with "in mass".

Point 4: In the Introduction "wild types presents two to..." ?? I think you mean: "wild types present from two to..." Plese, replace this.

Point 5: In the Itroduction "Foliar applicaton of nutrients..." in this sentence, remove "in" before "maximize" and then specify where the concentrations of micronutrients is maximized, please.

Point 6: In Material and Methods in the 2.1 paragraph the reference to "Figure 1" lacks twice. Please, insert them.

Point 7: In Material and Methods in the 2.2 paragraph the sentence about "the removalof weeding" is repeated twice. Please, remove one of them.

Point 8: When you write in Material and Methods about "sulfuric acid" and "sodium hydroxide", later you write them again but using their symbols. Please, choose to write the full names or the symbols for both times.

Point 9: In the Statistical analysis, it could be better to divide the sentence in two parts (i.e., after "their interactions"), because it is too long.

Point 10: In the Results the reference to "Table 1"misses in the text;

In the caption of" Table 5", "Grain" before "Yield" misses;

The reference to "Figure 2" misses in the text;

Add them in the text.

Point 11: In the Results, when you are describing "Table 6" the sentence "were statistically significantly their value range were higher." ...is not clear!! Could something be missing? Check it, please.

Point 12: In the Results the sentence from "Results show that crude protein" to "respectively" misses "only in the second year". Insert this, please.

Point 13: In the Results the comment about "Figure 3 (i), (ii), (iii)" and its reference are not in the text. You have to say something about this figure.

Point 14: In the Discussio, remove "that" between "who illustrated" and "the decrease of grain...".

Point 15: In the Discussion, when you talk about "The interaction among plants nutrients",  the last part of 4.1 paragraph should be reformulated and divided in two parts.

In this part, you shoud cite two clear examples of  nutrient interaction one specific for Strategy I plants: "Zuchi et al., 2015", and another specific for Strategy II plants: "Ciaffi et al., 2013".

Point 16: In the Supplementary Materials in the text, remove the reference of the "Table S1", since you have moved now the" Table 3 and 4" from the text to the Supplementary Materials.

Point 17: In general the text shows some mistakes of oversights (commas and dots). Please, read carefully it and check some double spaces. Moreover, e.g. "China" must be written with capital letter.

Iron and Zn after a dot must be written in full name.

Check "Triticum aestivum" that must be written in italics.

Author Response

Response to Reviewer 2 Comments (Minor revision)

Manuscript title: Winter Wheat Grain Quality, Zinc and Iron Concentration Affected by a Combined Foliar Spray of Zinc and Iron Fertilizers

Response to comments:

All comments provided by reviewer were addressed and given that most of them were minor errors, they were corrected and highlighted in the revised version of the manuscript and can be easily traced by using Trace change tool under Review menu of the MS word.

Point 1: The Introduction does not include relevant references regarding the topic of "agronomic biofortication technologies". In particular, not only nitrogen but also sulfur is important to facilitate the uptake of Fe and Zn. To this purpose, I suggest you to mention two papers: (1) Celletti et al., 2016 and (2) Astolfi et al., 2018, in which you can read about Fe accumulation in graminaceous plants and in their grains, that is affected by adequate and excess sulfur supply.

Response: This part of the introduction was revised after extensive reading on Sulfur interaction in Fe and Zn metabolism in plants. Sulfur as an important player was added to the literature. 

Point 2: In the first line of Introduction, replace"produced" with "cultivated", please.

Response: Corrected

Point 3: I recommend you to change some words that you were wrong to write in the Introduction:

"Foot" with "food";

"en masse" with "in mass".

Response: Corrected

Point 4: In the Introduction "wild types presents two to..." ?? I think you mean: "wild types present from two to..." Plese, replace this.

Response: Corrected

Point 5: In the Itroduction "Foliar applicaton of nutrients..." in this sentence, remove "in" before "maximize" and then specify where the concentrations of micronutrients is maximized, please.

Response: Corrected

Point 6: In Material and Methods in the 2.1 paragraph the reference to "Figure 1" lacks twice. Please, insert them.

Response: Corrected

Point 7: In Material and Methods in the 2.2 paragraph the sentence about "the removalof weeding" is repeated twice. Please, remove one of them.

Response: Corrected

Point 8: When you write in Material and Methods about "sulfuric acid" and "sodium hydroxide", later you write them again but using their symbols. Please, choose to write the full names or the symbols for both times.

Response: Corrected

Point 9: In the Statistical analysis, it could be better to divide the sentence in two parts (i.e., after "their interactions"), because it is too long.

Response: Revised

Point 10: In the Results the reference to "Table 1"misses in the text;

In the caption of" Table 5", "Grain" before "Yield" misses;

The reference to "Figure 2" misses in the text;

Add them in the text.

Response: Revised and added

Point 11: In the Results, when you are describing "Table 6" the sentence "were statistically significantly their value range were higher." ...is not clear!! Could something be missing? Check it, please.

Response: Revised

Point 12: In the Results the sentence from "Results show that crude protein" to "respectively" misses "only in the second year". Insert this, please.

Response: Revised and added

Point 13: In the Results the comment about "Figure 3 (i), (ii), (iii)" and its reference are not in the text. You have to say something about this figure.

Response: Revised and added

Point 14: In the Discussio, remove "that" between "who illustrated" and "the decrease of grain...".

Response: Corrected

Point 15: In the Discussion, when you talk about "The interaction among plants nutrients",  the last part of 4.1 paragraph should be reformulated and divided in two parts.

In this part, you shoud cite two clear examples of  nutrient interaction one specific for Strategy I plants: "Zuchi et al., 2015", and another specific for Strategy II plants: "Ciaffi et al., 2013".

Response: This part of the discussion was revised and more references were given including the proposed ones.

Point 16: In the Supplementary Materials in the text, remove the reference of the "Table S1", since you have moved now the" Table 3 and 4" from the text to the Supplementary Materials.

Response: revised and tables were renamed to give a better order to the numbers, Therefore, Table 1 remained table 1 , Table 2 remained table 2, Table 5 was renamed table 3, Table 6 was renamed table 4 and Table 7 was renamed table 5. Table 3 and 4 of the supplementary data were renamed Table S1 and Table S2, respectively.

Point 17: In general the text shows some mistakes of oversights (commas and dots). Please, read carefully it and check some double spaces. Moreover, e.g. "China" must be written with capital letter.

Iron and Zn after a dot must be written in full name.

Check "Triticum aestivum" that must be written in italics.

Response: Revised and checked

This manuscript is a resubmission of an earlier submission. The following is a list of the peer review reports and author responses from that submission.

Round  1

Reviewer 1 Report

The manuscript reports the effect of combined Zn and Fe applied as foliar fertilizer on winter wheat on grain quality, Zn and Fe concentration in grains.

The subject of the manuscript falls within the general scope of the Journal and the huge number of determinations in this experiment is relevant and worthy of consideration.

Increasing Fe and Zn concentration in wheat is an important global challenge due to high incidence of Fe and Zn deficiency in human populations. 

Major points:

·       Some authors (Aciksoz et al. 2011. Plant and Soil 349: 215–25), reports that the plant N status deserves special attention in biofortification of food crops with Fe.

·       I therefore consider it essential to describe with greater care the cultivation technique adopted (N, P and K fertilizer, water supply, rainfull, more information about temperature during the trials, etc.).

·       It is also necessary to define how the land subject of the study is classified (i.e. Low - Medium - High for Zn content; Low - Medium - High for Fe content). See also (if necessary) http://aes.missouri.edu/pfcs/soiltest.pdf

·       Another question is related to define the treatments compared: It is also necessary to define (on a scientific basis) how the three levels of Fe and Zn (13 - 9.5 and 5 kg) were chosen.

·       It would be preferable to find a simpler way to define the treatments compared in the experiments:

For example:

100%Zn13 = 3,0 kg ha-1 of Zn

100%Zn9.5 =   2,2 kg ha-1 of Zn

100%Zn5.5 = 1,3 kg ha-1 of Zn

80%Zn+20%Fe13 = 2.4 kg ha-1 of Zn + 0.52 kg ha-1 of Fe

80%Zn+20%Fe9.5 = 2.4 kg ha-1 of Zn + 0.38 kg ha-1 of Fe

and so on....

100%Fe13 =        2, 6 kg ha-1 of Fe

100%Fe9.5 =  1,9 kg kg ha-1 of Fe

100%Fe5.5 =       1,1 kg kg ha-1 of Fe

The discussion of the results must be improved and aimed at the objective of the work.

Then, the manuscript has several major limits and according to my opinion it should not be accepted in its current form. Important amendments are required before acceptance.  Therefore, I invite the authors to resubmission for a new review after major revisions.

Further comments and details are provided in the attached document.

Response to Reviewer 1 Comments

The manuscript reports the effect of combined Zn and Fe applied as foliar fertilizer on winter wheat on grain quality, Zn and Fe concentration in grains.

The subject of the manuscript falls within the general scope of the Journal and the huge number of determinations in this experiment is relevant and worthy of consideration.

Increasing Fe and Zn concentration in wheat is an important global challenge due to high incidence of Fe and Zn deficiency in human populations. 

Major points:

·Point 1: Some authors (Aciksoz et al. 2011. Plant and Soil 349: 215–25), reports that the plant N status deserves special attention in biofortification of food crops with Fe.

Response 1: It is true that N should be given a special attention in biofortification of food crops. In the introduction, to highlight this important subject, literature related to this was added.

·Point 2: I therefore consider it essential to describe with greater care the cultivation technique adopted (N, P and K fertilizer, water supply, rainfull, more information about temperature during the trials,  etc.).

Response 2: Cultivation technique description was revised, updated, and mentioned information were provided. Some information were missing in the submitted manuscript.

Point 3: It is also necessary to define how the land subject of the study is classified (i.e. Low - Medium - High for Zn content; Low - Medium - High for Fe content). See also (if necessary) http://aes.missouri.edu/pfcs/soiltest.pdf

Response 3: The document was consulted and the soil used was classified according to the classification in the document.

Point 4: Another question is related to define the treatments compared: It is also necessary to define (on a scientific basis) how the three levels of Fe and Zn (13 - 9.5 and 5 kg) were chosen.

Response 4: Three levels of Fe and Zn used in this experiment were chosen according the commonly used quantities of Zn and Fe fertilizers. It is observed that higher quantities of Fe and Zn were used in this study. The cultivar used in this study is a newly developed cultivar that has a strong resistance to cold, drought and diseases. Therefore, these quantities were selected and applied to test its resistance to different quantities of sprayed fertilizers.   

Point 5: It would be preferable to find a simpler way to define the treatments compared in the experiments:

For example:

100%Zn13 = 3,0 kg ha-1 of Zn

100%Zn9.5 =   2,2 kg ha-1 of Zn

100%Zn5.5 = 1,3 kg ha-1 of Zn

80%Zn+20%Fe13 = 2.4 kg ha-1 of Zn + 0.52 kg ha-1 of Fe

80%Zn+20%Fe9.5 = 2.4 kg ha-1 of Zn + 0.38 kg ha-1 of Fe

and so on....

100%Fe13 =        2, 6 kg ha-1 of Fe

100%Fe9.5 =  1,9 kg kg ha-1 of Fe

100%Fe5.5 =       1,1 kg kg ha-1 of Fe

Response 5: It is true that the way treatments were defined in the submitted manuscript was not showing the real quantities of Zn and Fe applied. This was corrected and the way of defining treatment proposed was adopted. However, names of treatments remained unchanged.

Point 6: The discussion of the results must be improved and aimed at the objective of the work.

 Response 6: The discussion was improved, deepened and linked with the objectives of the work accordingly

Point 7: Further comments and details are provided in the attached document.

Response 7: All 18 comments given in the attached documents were revised and addressed accordingly.  Corresponding errors, missing values, and texts were corrected/added to the manuscript at the right mentioned place in the text.

NOTE:   As recommended, yield data were double-checked from hard papers used recording data and first data entry and a transcription error was traced. Data for 20%Zn+80%Fe13 treatment were miss typed. Real data were used in the revised manuscript, which affected the text on interpretation of grain yield results and its discussion part.

Then, the manuscript has several major limits and according to my opinion, it should not be accepted in its current form. Important amendments are required before acceptance.  Therefore, I invite the authors to resubmission for a new review after major revisions.

Reviewer 2 Report

The currents study provided little pieces of information on the application of Zn and Fe in plant culture. In fact that the experimental design is simple, the discussion is general and superficial.

Response to Reviewer 2 Comments

Point 1: The currents study provided little pieces of information on the application of Zn and Fe in plant culture. In fact that the experimental design is simple, the discussion is general and superficial

Response 1: The current study was designed to study the effect of Fe and Zn on wheat as foliar fertilizers. The experimental design consisted of split-plots in a randomized complete block design with three replications, we believed that it could help in getting more details about the subject matter .  It is true that the introduction and discussion in the submitted manuscript were general; in the revised version of the manuscript, the discussion and introduction were revised and extended.

Reviewer 3 Report

I strongly suggest to extend the part of Introduction, including more references about the topic of nutritional quality of foods, biofortification of staple crops (as wheat), and about the crosstalk between nutrients, especially their effect when their content is deficient in the grains.

In the Introduction I suggest also that the authors have to be more precise when they talk about the "developing countries": "Which are the developing countries?".

And the "number of individuals", who suffer from zinc (Zn) and iron (Fe) deficiency, because "two billion individuals suffer from iron deficiency" And how many individuals suffer from zinc deficiency?.

In the Materials and Methods, precisely in the 2.3 paragraph, the references about the method to determine the "crude fat" is missing.

In the Results there is much confusion in their explanation. Pay attention because there are sentences that are repeated such as the authors comment about "crude fat of table 4" and "100%Fe fertilizers ratios of table 6".

All the tables must be standardized among them such as for example table 3 with table 4.

See table 2 in which "crude fiber" is missing.

Moreover, in the sentence of Results: " fertilizer quantity affected the spike length and the Zn content "crude fiber" is missing again.

I suggest to replace "seconded" with "followed" in the Results.

In the Discussion, in the 4.1 paragraph, the initial sentence is not well explained and I suggest a reference when the authors talk about: "the interaction can modify plant growth and yield".

In the Discussion, in the 4.3 paragraph, I suggest a reference when the authors talk about: "the accumulation of Zn and Fe in aleurone, endosperm and embryo...".

In the Discussion, in the 4.4 paragraph, from: "The general" to "significantly" the concept is repeated; please, modify this.

In general, the text shows several grammatical mistakes, please, check them.

I suggest to write only at the beginning of the paper "Zinc" and "Iron" in full with their abbreviation in the brackets, and then to use their abbreviation for all of the rest of the paper.

Response to Reviewer 3 Comments

Point 1: I strongly suggest to extend the part of Introduction, including more references about the topic of nutritional quality of foods, biofortification of staple crops (as wheat), and about the crosstalk between nutrients, especially their effect when their content is deficient in the grains.

Response 1: The introduction was revised and extended with much attention on the interaction of N, Fe, and Zn and the consequences of Zn and Fe deficiency in grains to human health.

Point 2: In the Introduction I suggest also that the authors have to be more precise when they talk about the "developing countries": "Which are the developing countries?".

Response 2: When the author used the term “developing countries”, was referring to the classification of countries from World Bank and the reference was provided in the revised manuscript.

Point 3: And the "number of individuals", who suffer from zinc (Zn) and iron (Fe) deficiency, because "two billion individuals suffer from iron deficiency" And how many individuals suffer from zinc deficiency?.

Response 3: Statistics were crosschecked and the number of individuals who suffer from Zn deficiency was given and references provided.

Point 4: In the Materials and Methods, precisely in the 2.3 paragraph, the references about the method to determine the "crude fat" is missing.

Response 4: The reference was provided in the reviced manuscript and the description of the method was detailed.

Point 5: In the Results there is much confusion in their explanation. Pay attention because there are sentences that are repeated such as the authors comment about "crude fat of table 4" and "100%Fe fertilizers ratios of table 6".

Response 5: Table 3 and 4 and their interpretations in the manuscript were redundant, therefore they were removed from the main text and placed in the Supplementary Materials. The text was revised due to a mistake found in the data entry that has affected the yield values of 20%Zn+80%Fe13 treatment.

Point 6: All the tables must be standardized among them such as for example table 3 with table 4.

Response 6: These tables were removed from the main text and sent to supplementary materials.

Point 7: See table 2 in which "crude fiber" is missing.

Response 8: This was corrected and Crude fiber added where it was missing.

Point 9: Moreover, in the sentence of Results: " fertilizer quantity affected the spike length and the Zn content "crude fiber" is missing again.

Response 9: The sentence was revised and crude fiber inserted .

Point 10: I suggest to replace "seconded" with "followed" in the Results.

Response 10: Seconded was replaced by followed as suggested .

Point 11: In the Discussion, in the 4.1 paragraph, the initial sentence is not well explained and I suggest a reference when the authors talk about: "the interaction can modify plant growth and yield".

Response 11: The reference was given

Point 12: In the Discussion, in the 4.3 paragraph, I suggest a reference when the authors talk about: "the accumulation of Zn and Fe in aleurone, endosperm and embryo...".

Response 12: The reference was given

Point 13: In the Discussion, in the 4.4 paragraph, from: "The general" to "significantly" the concept is repeated; please, modify this.

Response 13: These repeated words were removed

Point 14: In general, the text shows several grammatical mistakes, please, check them.

Response 14: The articles has undergone English language editing by MDPI. The text has been checked for correct use of grammar and common technical terms, and edited to a level suitable for reporting research in a scholarly journal.

Point 15: I suggest to write only at the beginning of the paper "Zinc" and "Iron" in full with their abbreviation in the brackets, and then to use their abbreviation for all of the rest of the paper.

Response 15: This was addressed; the full name was left in the title of the article